

# Continuous CO$_2$/CH$_4$/CO measurements (2012-2014) at Beromünster tall tower station in Switzerland

E. Satar[1, 2], T. A. Berhanu[1, 2], D. Brunner[3], S. Henne[3], and M. Leuenberger[1, 2]

[1] Climate and Environmental Physics, Physics Institute, University of Bern, Bern, Switzerland
[2] Oeschger Center for Climate Change Research, University of Bern, Bern, Switzerland
[3] Empa, Laboratory for Air Pollution/Environmental Technology, Dübendorf, Switzerland

*Correspondence to:* E. Satar (satar@climate.unibe.ch)

**Abstract.** The understanding of the continental carbon budget is essential to predict future climate change. In order to quantify CO$_2$ and CH$_4$ fluxes at the regional scale, a measurement system was installed at the former radio tower in Beromünster as a part of the Swiss greenhouse gas monitoring network (CarboCount CH). We have been measuring the mixing ratios of CO$_2$, CH$_4$ and CO on this tower with sample inlets at 12.5, 44.6, 71.5, 131.6 and 212.5 m above ground level using a cavity ring down spectroscopy (CRDS) analyzer. The first two-year (December 2012-December 2014) continuous atmospheric record was analyzed for seasonal and diurnal variations and inter-species correlations. In addition, storage fluxes were calculated from the hourly profiles along the tower. The atmospheric growth rates from 2013 to 2014 determined from this two-year dataset were 1.78 ppm yr$^{-1}$, 9.66 ppb yr$^{-1}$ and -1.27 ppb yr$^{-1}$ for CO$_2$, CH$_4$ and CO, respectively. After detrending, clear seasonal cycles were detected for CO$_2$ and CO, whereas CH$_4$ showed a stable baseline suggesting a net balance between sources and sinks over the course of the year. CO and CO$_2$ were strongly correlated ($r^2 >$ 0.75) in winter (DJF), but almost uncorrelated in summer. In winter, anthropogenic emissions dominate the biospheric CO$_2$ fluxes and the variations in mixing ratios are large due to reduced vertical mixing. The diurnal variations of all species showed distinct cycles in spring and summer, with the lowest sampling level showing the most pronounced diurnal amplitudes. The storage flux estimates exhibited reasonable diurnal shapes for CO$_2$, but underestimated the strength of the surface sinks during daytime. This seems plausible, keeping in mind that we were only able to calculate the storage fluxes along the profile of the tower but not the flux into or out of this profile, since no Eddy covariance flux measurements were taken at the top of the tower.

## 1 Introduction

Since industrialization, greenhouse gases in the atmosphere have been continuously increasing. According to IPCC (2013), CO$_2$ has increased from 278 ppm around the year 1750 to 390.5 in 2011, whereas CH$_4$ has increased from 722 ppb to 1803 ppb over the same period. The increase in atmospheric mixing ratios of CO$_2$ corresponds to less than half of the anthropogenic emissions from fossil fuels and land use change. The remaining fraction is removed from the atmosphere by sink mechanisms and stored in carbon reservoirs (Ciais et al., 2013; Le Quéré et al., 2015). Quantifying and understanding



these sinks is essential to improve our capabilities to predict future climate change. The ocean sink is relatively well understood, whereas the global land sink is difficult to quantify directly and therefore is often calculated as a residual from better constrained quantities, i.e. from fossil fuel consumption statistics, atmospheric growth rates, the ocean sink, and land use change estimates (Ciais et al., 2013; Le Quéré et al., 2015).

In order to fill in the gaps of knowledge in the carbon cycle and unravel the terrestrial sink, atmospheric observations of greenhouse gases are of prime importance. For instance, atmospheric inverse modeling enables the estimation of surface-atmosphere fluxes from measured atmospheric mixing ratios (Fan et al., 1998; Gurney et al., 2002; Rödenbeck et al., 2003; Broquet et al., 2013). However, traditionally, atmospheric measurements have been conducted in remote areas (Pales and Keeling, 1965) to reduce the influence of nearby sources. The measured mixing ratios of greenhouse gases at such

background sites provide information about the well mixed atmosphere at the hemispheric scale (Gloor et al., 2000). Two main limitations arise from this selection of sites: the resulting flux estimates are only representative for large scales and the measurements are biased towards the marine boundary layer or the free troposphere. In order to improve the understanding of land-atmosphere fluxes at regional scales, a greater share of data from continental sites is essential (Gloor et al., 2000). Tans (1993) proposed several ways of monitoring the carbon cycle on continents including tall tower measurements.

Observations from continental sites may take place in an environment of complex sink, source, and transport processes (due to local effects), and may thus be characterized by strong variability on different time scales. However, both inverse-modeling (Peylin et al., 2005) and budget calculation approaches from atmospheric mixing ratios of greenhouse gases (Bakwin et al., 2004) will be better constrained with the availability of high frequency continuous records.

Atmospheric tall tower observations have a relatively short history in contrast to flask samplings at remote sites, which span

more than 20 years. The first tall tower atmospheric measurements were launched in the United States in the early 1990s (Bakwin et al., 1995; Bakwin et al., 1998) were followed by atmospheric tall towers in Europe and Asia (Popa et al., 2010; Winderlich et al., 2010; Thompson et al., 2009; Schmidt et al., 2014; Vermeulen et al., 2011). In the early 21[st] century, a European network of tall tower stations was established under the umbrella of the CHIOTTO (Continuous High Precision Tall Tower Observations of Greenhouse Gases, www.chiotto.org) project. This tall tower network is currently being

integrated into a much broader European carbon observation system named ICOS (Integrated Carbon Observation System, www.icos-infrastructure.eu).

In Switzerland, a greenhouse gas observation network with four new measurement sites was established in the framework of the project CarboCount CH (Oney et al., 2015) with a main objective of quantifying local to regional fluxes of greenhouse gases by combining top-down and bottom-up approaches. An important component and the only tall tower site of this

network is the former radio tower at Beromünster (47° 11′ 23″ N, 8° 10' 32″ E, 212.5 m tall, base at 797 m a.s.l), which is located in a moderately hilly environment at the southern border of the Swiss Plateau, the most densely populated and agriculturally used region of Switzerland between the Alps in the south and the Jura mountains in the northwest.





Here we present the first two years (December 2012-December 2014) of continuous measurements of $CO_2$, $CH_4$ and CO at five different heights from this tall tower. We investigate the seasonal and diurnal variations of the three species, the development of vertical gradients between the different heights and the corresponding storage fluxes accumulated below the sampling height, and we draw inferences about the correlations between species. With this purely observation-based multi-species and multi-level approach, we characterize the site Beromünster with respect to the influences of different sinks (photosynthesis ($CO_2$), OH reaction (CO)) and sources (respiration ($CO_2$), fossil fuel combustion ($CO_2$, CO) and ruminants ($CH_4$)) of local and regional origin at diurnal to annual time scales. Combining observations with modeling approaches is done as part of additional publications using the CarboCount CH measurements (Oney et al., 2015; Henne et al., submitted).

## 2 Methods

### 2.1 Measurement system

The measurement system at the Beromünster tall tower was installed in October 2012, and mixing ratios of $CO_2$, $CH_4$, CO and $H_2O$ have been measured continuously since, using a Picarro CRDS analyzer (G-2401). The technical setup of the measurement system as well as the data processing concept and calibration strategy were explained in detail in Berhanu et al.(2015). Here we provide a brief summary of the measurements: Sampling was conducted from five different elevations, 12.5, 44.6, 71.5, 131.6 and 212.5 m above ground level and at each level, the ambient air is measured for three minutes. From these three-minute measurements, only the last 60 seconds are used in the data analysis to prevent contamination from the previous measurement. In addition to the measurements of trace gases, meteorological data, including wind speed, wind direction, air temperature, barometric pressure, and humidity, are measured at each height.

### 2.2 Time series analysis

The operation cycle described above resulted in time series of four measurements per hour at each of the five elevations, with each measurement representing a one-minute average. The two years of data were analyzed for seasonal and diurnal variations, for correlations between species, and to estimate greenhouse gas fluxes.

For the analysis of seasonality, first we estimated a smoothly varying background based on the measurements at the highest elevation using the complete 25-month record. In order to ensure that the background values are not affected by local sources and sinks, we filtered the time series for pollution or depletion events using an iterative approach. First, a moving average of 30 days was calculated and the standard deviation ($\sigma$) of the differences from this moving average was computed. Subsequently, all measurements with differences exceeding 2-$\sigma$ were eliminated, and a new running mean was computed. This procedure was repeated until only points within the 2-$\sigma$ limits were retained. Then, to estimate the trend of the background values, a function composed of 2 seasonal harmonics and a linear term (Thoning et al., 1989; Masarie and Tans,





1995) was fitted to the two full years of data (Eq. (1)). Finally, the atmospheric record was detrended using the calculated trend ($a_1$).

$$f(t) = a_0 + a_1 t + \sum_{n=1}^{2} \left( b_n \sin \frac{2\pi n t}{365} + c_n \cos \frac{2\pi n t}{365} \right)$$

(1)

where, $t$ is time in days, and $a_0$, $a_1$, $b_n$ and $c_n$ are the coefficients of the fit.

In addition to calculating a background based on the iterative approach described above, we have also applied the robust

extraction of baseline signal (REBS) technique (Ruckstuhl et al., 2012) for comparison. This baseline estimate is a statistical method using non-parametric local regression with robustness weights. By default, the method assumes an asymmetric contamination of the baseline signal due to local pollution events. Therefore, in REBS, the left-hand side of the residual distribution is used to estimate the standard deviation of the baseline, which should approximately follow a Gaussian behavior. However, in the presence of terrestrial sinks and fossil fuel sources of $CO_2$, excursions below and above

background levels are measured. Therefore, a symmetric estimator of the standard deviation was applied in the case of $CO_2$. REBS was applied to all species with a bandwidth of 60 days and a maximum of 20 iterations.

For the analysis of correlations between species, a moving average of five days was used to eliminate slow variations including seasonal variability and to concentrate on short term fluctuations, in particular on diurnal variability and on the time scales of pollution events which typically lasted from a few hours to a few days. Since both variables used in the

regression analysis are subject to uncertainty, standard major axis regression (Legendre and Legendre, 2012) was applied to the residual time series.

For the analysis of diurnal cycles and for the estimation of fluxes, only data starting from January 2013 was used to ensure an equal share of data from each month. Monthly mean diurnal cycles were calculated by averaging the corresponding hourly values within a month for each elevation. Five percent of the highest and lowest measurements per month and hour

were trimmed to exclude extreme values. The data was detrended prior to averaging, in order to eliminate bias resulting from data gaps. For the estimation of fluxes, hourly mean mixing ratios were used. The approach used for the flux estimation is explained further in the following section.

### 2.3 Flux estimation

The net ecosystem exchange (NEE) is the net flux into or out of the ecosystem, and can be calculated from atmospheric

measurements as the sum of the advective flux, the storage flux and the turbulent flux (Lee, 1998; Finnigan, 1999; Yi et al., 2000). Here, the method presented by Winderlich et al. (2014) is adopted as it was specifically developed for measurements from multiple levels along a tall tower. Similar to previous studies (Haszpra et al., 2005; Winderlich et al., 2014) the advective flux component was not included in the calculations as this information was not available. Advective fluxes were shown to depend largely on topography and site characteristics (Aubinet et al., 2005; Feigenwinter et al., 2008) with an




estimated contribution of about 10% of the overall flux for a very tall tower (Yi et al., 2000), but may occasionally be the dominating component. However, these fluxes are difficult to measure and vertical advection is likely to be compensated for by horizontal advection (Feigenwinter et al., 2004). Due to the lack of direct eddy flux measurements at the tower, a turbulence term could also not be taken into account. Thus, in our calculations, the flux estimates are restricted to the storage flux term.

The storage flux ($F_{storage}$) between the surface and the top level $z_r$ of the tall tower at a given time $t$ can be calculated from the gradients in mixing ratios along the tower according to:

$$F_{storage}(t, z_r) = \int_0^{z_r} \frac{1}{V_m(z)} \cdot \frac{\partial c(t,z)}{\partial t} \cdot dz \qquad (2)$$

where $t$ is time, $V_m$ is the molar volume of air, and $c$ is the mixing ratio of the measured trace gas. In order to solve the integral in Eq. (2), Winderlich et al. (2014) suggested to discretize the problem. The integral was approximated by trapezoidal areas, where concentrations are linearly interpolated between subsequent levels:

$$F_{storage}(t, z_r) = \sum_{h=1}^4 \frac{\frac{1}{2}(\rho_h + \rho_{h+1})}{m_{air}} \cdot \frac{\frac{1}{2}\left((c_h(t_{i+1}) - c_h(t_i)) + (c_{h+1}(t_{i+1}) - c_{h+1}(t_i))\right)}{t_{i+1} - t_i} \cdot (z_h - z_{h+1}) \qquad (3)$$

where $\rho$ is the air density, $m_{air}$ is the molar mass of air, and $z_h$ is the sampling height ($z_1$=212.5 m, $z_2$= 131.6 m, $z_3$=71.5 m, $z_4$=44.6 m and $z_5$=12.5 m). The fluxes were individually calculated for each consecutive height interval for a time step of one hour. The total storage flux for each time step was obtained by taking the sum of the four flux estimates for the different height intervals. Note that in the absence of advective and turbulent fluxes a positive storage flux indicates a net source, whereas a negative flux implies a sink.

Calculations were done on a daily basis and averaged over a month, resulting in monthly averaged daily fluxes. The uncertainties were calculated using the standard error of the mean, considering the number of days as the sample size.

## 3 Results and Discussion

### 3.1 Seasonal variations and annual growth rate

Figure 1 shows the mixing ratios of the trace gases from the highest elevation. Points outside of the blue band are considered to be either pollution or depletion events due to local and regional sources and sinks. The 2-σ filter applied in the estimation of the background values eliminated 31%, 34%, and 33% of the measurements for $CO_2$, $CH_4$ and CO, respectively.

$CO_2$ shows a clear seasonal cycle, with a mean peak-to-peak amplitude of 13.1 ppm for the two-year period (Fig. 1a). The seasonal variations of $CO_2$ are associated with biological activity; the minimum occurs in August and the maximum in



March, as expected for a northern hemisphere site (Randerson et al., 1997). The observed amplitude can be compared with other tall tower stations in Europe: For Cabauw in the Netherlands (Vermeulen et al., 2011), Bialystok in Poland (Popa et al., 2010), and Ochsenkopf in Germany (Thompson et al., 2009), the observed seasonal amplitudes were 25, 20 and 15.5 ppm, respectively. The observed seasonal variation at Beromünster is comparatively low, but this may partly be explained by the

filtering of the data. Other studies have not specifically filtered out the pollution events but took trimmed (means from the inter-quartile (25%-75%) range) mean values. Excursions from the baseline would tend to be positive in winter and negative in summer, yielding overall greater amplitudes of the monthly means in non-filtered data. Applying the latter estimation, a larger fraction of the regional signal would still remain in the baseline. At Beromünster, taking the daily means from the inter-quartile (25%-75%) range results in a mean peak-to-peak amplitude of 18.2 ppm over the 2 years.

The $CH_4$ mixing ratios (Fig. 1b) varied between 1900 and 2200 ppb. Most of the variability is due to pollution events with mixing ratios varying by up to a few hundred ppb over time periods from minutes to days. The short-term spikes are likely due to emissions from agricultural activities in the vicinity of the tower (mostly ruminants), while the longer lasting peaks are related to synoptic variability of atmospheric transport and mixing. After eliminating such pollution events, the detrended time series exhibited a rather constant baseline with only a weak maximum in early spring. In Switzerland, more than 80% of

the methane emissions are due to agriculture, mainly from ruminants (Swiss Federal Office for the Environment FOEN, 2015a), and the emissions are expected to be larger in spring and summer months due to higher temperatures in the manure storage and larger productivity of the dairy cows (Henne et al., submitted). Superimposed on the variability of the sources is the seasonality of the OH radical, which is the major $CH_4$ sink. The destruction of $CH_4$ by OH is most pronounced during summer months causing a $CH_4$ minimum in the northern hemisphere in late summer (Dlugokencky et al., 1994). At

Beromünster, the hemispheric background signal and the regional scale emissions most likely cancel each other yielding a rather constant baseline estimate. Therefore, seasonal amplitudes for $CH_4$ were not calculated. In contrast to Beromünster, other tall tower stations in Europe (Thompson et al., 2009; Popa et al., 2010; Schmidt et al., 2014; Vermeulen et al., 2011) have reported more pronounced seasonal amplitudes between 35-70 ppb.

For CO (Fig.1c), higher values (up to 450 ppb) were observed in winter, and lower values (around 100 ppb) were observed

in summer. The seasonal variations in CO are largely governed by anthropogenic emissions and the strength of the OH sink. The main sources of CO in Switzerland are the transport sector and residential heating, corresponding to 53% and 37% of the emissions, respectively (Swiss Federal Office for the Environment FOEN, 2015b). The observed seasonality might be associated with enhanced anthropogenic emissions of CO in winter months from increased heating and reduced atmospheric mixing. In addition to the variability of the sources, the main CO sink, which is the reaction of CO with OH radicals, also

varies seasonally with a maximum during summer months. This sink may be partially compensated by enhanced chemical production of CO through the oxidation of $CH_4$ and VOCs which is also largest in summer (Granier et al., 2000). The timing of the maximum of the seasonal variation is similar to $CO_2$, and the mean peak-to-peak amplitude of CO is 47.7 ppb averaged over the two years. Other tall tower stations in Europe reported much higher seasonal amplitudes between 75 and





130 ppb (Thompson et al., 2009; Vermeulen et al., 2011; Popa et al., 2010). Similar to $CO_2$, some of the differences in the observed seasonal amplitudes might be related to data selection, and taking the daily means from the inter-quartile (25%-75%) range results in higher mean peak-to-peak amplitude of 54.5 ppb over the 2 years. Nevertheless, the comparison suggests that the Beromünster site has a more rural character with a lower anthropogenic peak in winter than these other

sites. It should also be noted that CO emissions in Europe are still decreasing significantly (European Environment Agency EEA, 2014), which will likely be reflected in a corresponding decrease in the seasonal amplitudes of the above mentioned European tall tower sites.

The analysis of seasonality was also carried out for all sampling heights. The background estimates were calculated for each level individually, but the detrending was done according to the highest elevation since it has the lowest short-term

variability. The peak-to-peak amplitudes for $CO_2$ and CO were largest for the lowest elevation. This difference in amplitudes was related to higher maxima in winter rather than the lower minima in summer. The lowest summer minimum was observed at the topmost sampling height. This could be explained by the rectifier effect: during summer and spring months when photosynthesis is active and vertical mixing is strong, atmospheric $CO_2$ accumulates near the surface (Denning et al., 1999). Therefore, higher concentrations near the surface and lower concentrations aloft are observed.

The 2013-2014 growth rates obtained from the regression fit are given in Table 1. For $CO_2$ and $CH_4$, positive annual growth rates are calculated. The calculated growth rate of $CO_2$ (1.78 ppm yr$^{-1}$) was slightly lower than the global increase of 2.07 ppm yr$^{-1}$ during the last decade , whereas the growth rate of $CH_4$ was higher than the global increase of 3.8 ppb yr$^{-1}$ (World Meteorological Organization WMO Global Atmosphere Watch GAW, 2014). For CO, a decrease of -1.27 ppb yr$^{-1}$ was obtained, which is in agreement with the generally decreasing trend in CO mixing ratios in Europe (European Environment

Agency EEA, 2014; Swiss Federal Office for the Environment FOEN, 2014; Zellweger et al., 2009). However, the Beromünster time series is too short to determine reliable trend estimates, particularly in the presence of large inter-annual variability of meteorology (Warwick et al., 2002). This is especially true in the case of small relative trends and large local flux contributions (as in the case of $CH_4$).

In general, our seasonal amplitude estimates were lower for all species when compared with other tall tower sites. This may

partly be explained by the differences in the estimation of the background values: among different tall tower studies, the selection of the background mixing ratios also varied. Taking the measurements from the highest elevation is common in most studies in order to exclude local effects (Bakwin et al., 1995; Popa et al., 2010; Thompson et al., 2009; Winderlich et al., 2010; Vermeulen et al., 2011; Schmidt et al., 2014). Trimmed daily means (means from the 25%-75% range), daytime minima (Vermeulen et al., 2011) or the mean of afternoon hours (Popa et al., 2010), were taken in the mentioned studies.

However, with such approaches, our time series were still dominated by frequent stable atmospheric conditions that lead to emission accumulation and lasted up to a couple of days. As such events occur mostly in winter time, the maxima in the seasonal cycles were affected by their strength and the duration. Since the time series is relatively short, the inter-annual



variability of the seasonal amplitudes was also dominated by the frequency of these events. Therefore, we might have underestimated the seasonal amplitudes when eliminating the local events.

As an independent approach to our elimination of local events, we have also conducted the seasonality analysis by using the so-called robust extraction of baseline signal (REBS) technique (Ruckstuhl et al., 2012). The resulting baseline is indicated

by the green curve in Fig. 1. The two different baseline estimates for $CO_2$ and $CH_4$ closely follow each other, whereas CO exhibits a difference of 15 ppb in the maxima of 2013. This might be related to different responses of the two background filters to long lasting pollution events which were particularly prominent in the beginning of the dataset. The difference in the two baseline estimates resulted in a smaller negative annual growth rate of -0.20 ppb yr$^{-1}$ for REBS as compared to -1.27 ppb yr$^{-1}$ for the background filter. Since the time series is relatively short, both methods produce results that are still strongly

affected by inter-annual variability.

**3.2 Correlations between species**

Figure 2 shows the reduced major axis regression slopes (left panel), and coefficient of determination (right panel) for all months and all heights. Slopes are presented on a mass basis rather than a molar basis to enable a direct comparison with ratios expected from emission inventories. By using the residuals of 5-day moving averages, we expect to track short-term

changes which are most probably related to anthropogenic influences. However, even after the elimination of seasonality, both the regression slopes and the coefficients of determination showed seasonal variations. Winter months showed higher coefficients of determination ($r^2$) than summer months, with January being the highest: 0.92 and 0.80 for $\Delta CO/\Delta CO_2$ and $\Delta CO/\Delta CH_4$, respectively.

The strong seasonality seen in $\Delta CO/\Delta CO_2$ ratios is probably a result of the seasonality of the biospheric $CO_2$ fluxes. In

summer, the correlation between CO and $CO_2$ almost drops to zero, which makes the determination of the slopes very uncertain. Taking only the well mixed afternoon values slightly improves the correlation and results in somewhat higher slopes for the summer months. It is unclear to what extent the $\Delta CO/\Delta CO_2$ ratios in summer can be interpreted in terms of anthropogenic $\Delta CO/\Delta CO_2$ emission ratios and to what extent this ratio is influenced by biospheric $CO_2$ fluxes. For winter months, on the other hand, the very tight correlations suggest that biospheric $CO_2$ fluxes play only a minor role and that the

ratios are dominated by collocated anthropogenic emissions of CO and $CO_2$. The regression slopes for the winter months can therefore be compared with the most recent Swiss inventory estimates available for the year 2013 (Swiss Federal Office for the Environment FOEN, 2015a, b), indicated by the horizontal dashed lines (Fig. 2a). During winter months, the observed $\Delta CO/\Delta CO_2$ ratio is close to the ratio expected from the anthropogenic emission inventories, suggesting that the ratio in the inventories is consistent with atmospheric observations. However, the ratio observed at Beromünster in winter is still

somewhat lower than the annual mean emission ratio of 4.96 g/kg reported for the year 2013. Considering the negative trends of CO in Europe the agreement is reasonable. A comprehensive analysis of CO/$CO_2$ ratios including the use of radiocarbon data is part of a forthcoming publication (Berhanu et al., in preparation).



For $\Delta CO/\Delta CH_4$ ratios (Fig. 2c) there is reasonable agreement between the measurements and the inventory estimates (Swiss Federal Office for the Environment FOEN, 2015a, b). However, observed ratios were larger than the inventory annual mean ratio from December to March and were lower from May to October. The seasonality in the observation-based ratios is likely dominated by the seasonality in CO emissions, which are expected to peak during the cold season. Anthropogenic sources of

$CH_4$, conversely, are expected to be relatively constant over the year (Hiller et al., 2014), though there is a tendency to smaller emissions in the cold season (Henne et al., submitted).

Besides seasonality, there are also differences in the height dependence of the correlations. In contrast to the ratios $\Delta CO/\Delta CO_2$, the wintertime ratios $\Delta CO/\Delta CH_4$ showed a clear dependence on height, ranging from 1.48 g/g at the highest to 1.18 g/g at the lowest level (Fig. 2c). Since the atmospheric mixing properties should be independent of species, the different

height dependencies may be a reflection of differences in the relative importance of local versus distant sources. CO emissions from combustion processes are closely coupled to anthropogenic $CO_2$ emissions (Gamnitzer et al., 2006). Therefore, $\Delta CO/\Delta CO_2$ ratio along the tower is not likely to change because of a relatively large distance to sources of fossil fuel combustion. However, for emissions from agriculture, no direct link between $CH_4$ and CO exists. The wider scatter ($r^2 = 0.54$ for January) and the lower ratio of the emissions at the lowest level suggest larger than average local emissions of $CH_4$.

This is in agreement with the location of the tower in an area dominated by agricultural use (Hiller et al., 2014).

### 3.3 Diurnal variations

The monthly mean diurnal cycles of $CO_2$, CO and $CH_4$ for all sampling levels are shown in Fig. 3. Each x-axis spans 24 hours centered at noon. The times correspond to UTC, whereas the local time is UTC+1.

The diurnal cycles of $CO_2$ were very pronounced in the spring and summer months (Fig. 3a), and showed distinct differences

among the inlet heights. At lower heights the diurnal variations are amplified by local effects. Differences between the height levels were largest during night time, stable atmospheric conditions, when $CO_2$ emitted by plant respiration and anthropogenic emissions accumulate near the ground and only slowly mix to higher altitudes. $CO_2$ mixing ratios started to decrease with sunrise due to the uptake of $CO_2$ by plants and the simultaneous break-up of the nocturnal boundary layer by convective vertical mixing. The top of the growing planetary boundary layer (PBL) in the morning reached the highest inlet

with a delay of about 2 hours, as indicated by the time lag of the early morning peak between the highest and lowest inlet. Due to strong mixing in the fully developed convective PBL, the gradients along the tower disappeared around noon. The mixing ratios continued to decrease through the day until a minimum was reached in the late afternoon. At this point, the lowest mixing ratios were measured at 12.5 m, which likely reflects the influence of net uptake of $CO_2$ by photosynthetic activity in the surroundings of the tower. Vertical gradients started to reappear in the evening when convective mixing slows

down due to surface cooling. In contrast to the spring and summer months, winter months did not show distinct diurnal cycles, but a rather stable layering with distinctly different mixing ratios at the different sampling levels. During these months, the highest mixing ratios were always seen at the lowest level, suggesting that the surface is a net source of $CO_2$





during this time of the year. Throughout the course of the year, the amplitudes of the diurnal cycles varied, reaching a maximum amplitude of 17 ppm in July at the 12.5 m height

$CH_4$ observations (Fig. 3b) showed a similar diurnal pattern to $CO_2$ driven by atmospheric mixing processes and planetary boundary layer evolution. However, the early morning peak was lagging behind that of $CO_2$. Again, this indicates different

dominating source and sink mechanisms and locations. As mentioned in Sect. 3.3.1, the main source of methane in the vicinity of the tower and the whole of Switzerland are emissions from ruminants. These are expected to remain rather constant throughout a day/night cycle although recent studies on direct emissions from grazing ruminants suggest lower night-time emissions (Felber et al., 2015). In contrast to $CO_2$, the primary sink of methane is the destruction by OH radicals, which is expected to experience a diurnal cycle related to the production mechanism of the radicals (Ehhalt, 1999). However,

considering the very slow reaction of $CH_4$ with OH (lifetime 9-10 years) (Kirschke et al., 2013), no significant diurnal cycle resulting from chemical loss is expected. Relatively high mixing ratios were observed at the lowest inlet compared to other levels, indicating strong local sources. The latter can be directly related to cows that either graze in the vicinity of the tower or are housed in nearby farmsteads.

In contrast to $CO_2$ and $CH_4$, the diurnal variation of CO (Fig. 3c) is not clearly visible and much smaller than its annual

cycle. In winter months, when the diurnal evolution of the PBL is little pronounced and convective mixing is small, the diurnal variations were only weak and showed a pattern similar to $CO_2$ and $CH_4$. Highest and most variable CO mixing ratios were measured at the lowest level, whereas at the highest inlet, CO mixing ratios stayed relatively constant throughout the day. In the summer months, CO mixing ratios also showed pronounced diurnal cycles governed by atmospheric mixing processes and PBL evolution. Although the lifetime of CO is on the order of weeks in summer (Seinfeld, 2006), the effect of

the OH sink during the day most plausibly will not be seen, since transport and mixing processes occur on much shorter time scales.

The differences in the timing of the diurnal variations for $CO_2$, $CH_4$, and CO together with the cosine of the solar zenith angle (NOAA Solar Calculator, http://www.esrl.noaa.gov/gmd/grad/solcalc/) are shown in Fig. 4. At 06.00 in the morning, $CO_2$ mixing ratios start to decrease with the onset of photosynthesis, whereas $CH_4$ and CO accumulation continues in the still

shallow PBL. The decrease in the $CH_4$ mixing ratios starts later and only with the growth of the PBL. The decrease in the CO mixing ratios occurs even later, suggesting emissions in the morning, possibly from vehicles although there is not much traffic in the direct vicinity of the tower. The disappearance of the vertical gradient of $CO_2$ thus appears to be a combination of surface uptake and convective mixing, whereas the decreases in $CH_4$ and CO mixing ratios are determined mostly by vertical mixing.

**3.4 Flux estimation**

Our storage flux estimates are based on the vertical gradients along the tower. In order to illustrate the development of the gradients, monthly mean mixing ratios of $CO_2$ versus height were plotted for the five elevations along the tower and for





different hours of the day (Fig. 5). This was done for two selected months in summer and winter, respectively. For better visibility, only every third hour is plotted. In June, the monthly average mixing ratios varied over a broad range of 15 ppm over the course of the day, with pronounced vertical gradients at night, but little to no gradients during afternoon hours (Fig. 5a). In contrast, the mixing ratios in January varied over a much narrower range of five ppm, with gradients between different heights persisting throughout the day (Fig. 5b).

The areas enclosed between the profiles of two consecutive hourly time steps and heights correspond to the storage flux between those heights for that time interval. These fluxes were summed up to yield the total storage flux between the lowest and highest level.

According to Winderlich et al. (2014), this approach yields the most reliable estimates of NEE during nighttime when vertical mixing is reduced and NEE is dominated by the storage flux term. During daytime, vertical gradients of mixing ratios are much less pronounced due to vertical mixing and NEE is dominated by the turbulent flux through the top of the observed profile rather than the storage flux. Since this turbulent import or export of $CO_2$ into the vertical column through the highest level was not measured at Beromünster, our approach is likely to underestimate the surface sinks of $CO_2$ during daytime in the growing season. Moreover, the lowest sampling height in Beromünster tower is 12 m, thus, the variations in mixing ratios below 12 m could not be taken into consideration. This might as well lead to underestimation of the fluxes.

In Fig. 6, monthly averaged diurnal cycles of the storage fluxes for the years 2013 and 2014 are shown for all three measured species. The calculated flux estimates for the winter months were insignificantly different from zero for all species, and included large uncertainties. Although the atmosphere was usually stably stratified, the concentration increments between consecutive time steps were too low and the day-to-day variability was too large to yield reliable storage flux estimates. Hence, no conclusions could be drawn for winter months.

Between April and September, $CO_2$ (Fig. 6a) showed cycles of positive nighttime and negative daytime fluxes. Respiration of the biosphere is the most likely explanation for the positive fluxes at nighttime. The average night-time (23.00-04.00) storage flux estimate for these months was $1.57 \pm 0.11$ µmol m$^{-2}$ s$^{-1}$. With the onset of photosynthesis and turbulent vertical mixing, the storage fluxes changed to negative values in the morning and remained mostly negative until the late afternoon, though afternoon fluxes were lower than morning fluxes. The largest negative fluxes of about -6.8 µmol m$^{-2}$ s$^{-1}$ were reached in July. The average daytime (11.00-16.00) flux for the April - September time period was calculated as $-1.56 \pm 0.08$ µmol m$^{-2}$ s$^{-1}$. It should be noted that the day-time storage fluxes are likely to underestimate NEE.

The pattern obtained for $CH_4$ (Fig. 6b) was not as clear as in the case of $CO_2$. For the same period (April - September), near-zero nighttime storage fluxes and a maximum storage flux in the morning hours lagging behind the nighttime maximum in the $CO_2$ profiles was observed. An explanation for this early morning peak could be the combined effect of the $CH_4$ sources in the region, and the mixing properties of the atmosphere. $CH_4$ accumulates near the ground during nighttime and prior to the start of the vertical mixing in the early morning hours. The accumulated emissions reach the upper levels of the tower



only after the rise of the PBL, hence a time shifted maximum of fluxes occur. For $CH_4$, the nighttime (23.00-04.00) average storage flux estimate for the two years from April to September was calculated as $3.21 \pm 0.78$ nmol m$^{-2}$ s$^{-1}$, whereas for the early morning hours (04.00-08.00), the average storage flux estimate was $13.99 \pm 1.18$ nmol m$^{-2}$ s$^{-1}$.

In contrast to $CO_2$, where fluxes are expected to be influenced by photosynthetic activity, $CH_4$ is not related to a sink
mechanism on the time scales of concern. Therefore, it is very likely that the negative fluxes calculated for $CH_4$ during the afternoon hours are related to vertical mixing and should be offset by strongly positive turbulent fluxes at the top of the tower. Since the method was applied in full analogy, the negative fluxes for $CO_2$ during afternoon hours may also be significantly affected by turbulent fluxes into the column and not only by plant uptake.

In analogy to $CO_2$ and $CH_4$, flux estimates for CO were also calculated (Fig. 6c). However, the pattern is less clear, even
when looking at the April – September period, which is probably due to relatively small night-time CO emissions in the local environment, as was already indicated by the analysis of the CO diurnal cycle. Nevertheless the average flux of the early morning hours (04.00-08.00) was calculated as $2.10 \pm 0.26$ nmol m$^{-2}$ s$^{-1}$ for the summer months.

In addition to the estimation of average fluxes, the approach enabled us to capture some differences between the years. For example, the daily variations in storage fluxes in April 2014 showed stronger uptake when compared with the previous year.
This difference can be explained by the relatively cold spring in 2013 (MeteoSwiss, meteoswiss.ch).

The diurnal variations of fluxes in Beromünster are similar to Winderlich et al. (2014). For the summer months (June – September) of 2009 – 2011, the mean of the total nighttime (23.00-04.00) flux was reported as $2.7 \pm 1.1$ µmol m$^{-2}$ s$^{-1}$ and $5.6 \pm 4.5$ nmol m$^{-2}$ s$^{-1}$ for $CO_2$ and $CH_4$ respectively (Winderlich et al., 2014). In Beromünster taking the summer months (June – September) yielded a mean storage flux of $1.8 \pm 0.2$ µmol m$^{-2}$ s$^{-1}$ and $3.6 \pm 1.0$ nmol m$^{-2}$ s$^{-1}$ for $CO_2$ and $CH_4$ respectively.
There exists a reasonable agreement between these flux estimates.

In order to compare our estimates with other measurements of NEE in Switzerland, we calculated the flux sums for an entire year, keeping in mind that the estimates for winter months are small yet unreliable. For the years 2013 and 2014, the flux sums were -29 g C m$^{-2}$ and -35 g C m$^{-2}$, respectively. In Switzerland, ecosystem flux measurements are done within the Swiss FluxNet using the Eddy covariance technique (www.swissfluxnet.ch). Chamau and Früebüel are grassland sites about
18 and 30 km away from the Beromünster tall tower, respectively. NEE for the years 2006 and 2007 was reported as -222 g C m$^{-2}$ and -417 g C m$^{-2}$ in Früebüel, and -59 g C m$^{-2}$ and -69 g C m$^{-2}$ in Chamau, respectively (Zeeman et al., 2010). Our estimates for Beromünster seem to be reasonable considering that (i) the interannual variations in climatic conditions strongly affect the NEE; (ii) the sites are not representative of the same land cover; (iii) the storage flux estimate for winter months are highly uncertain and (iv) they may include contributions from anthropogenic $CO_2$ emissions throughout the year.
As explained earlier, this approach tends to underestimate NEE during daytime when it is expected to be most negative. Measurement of the turbulent fluxes at the top of the tower would be very valuable to close the budget. However, the presented storage fluxes can still provide a good first guess of the general temporal flux pattern if no turbulent and horizontal




flux data are available. In such a situation, a multi-species approach (tracer ratio method) can be valuable, since it allows – at least qualitatively – distinguishing imprints from vertical mixing and source/sink mechanisms.

## 4 Conclusions and Outlook

We have presented an extensive analysis of the first two years of $CO_2$, $CH_4$ and CO measurements at a new tall tower site in Switzerland. The data were analyzed for seasonal and diurnal variations, correlations between species, and storage fluxes within the atmospheric column below the highest elevation on the tower. Growth rates from 2013 to 2014, as well as mean seasonal cycles were estimated based on background mixing ratios determined for the three species. The correlations between species showed a strong link between CO and $CO_2$ in winter, but not in summer, suggesting that $CO_2$ variations were dominated by anthropogenic emissions in winter and by biospheric fluxes in summer, respectively. The diurnal profiles of the trace gases at the different sampling heights on the tower and in different seasons are largely controlled by diurnal variations of vertical mixing but also by local sources and sinks, particularly in the case of $CO_2$. Lastly, storage flux estimates showed pronounced daily variations and are believed to provide a reasonable estimate of the surface fluxes during night-time and morning hours, but are potentially different from surface fluxes later during the day due to increased turbulent fluxes at the top of analyzed column.

Although our data was limited to two years, it enabled a general characterization of the site Beromünster. The correlation between CO and $CO_2$ will be further investigated with the addition of radiocarbon measurements. Moreover, the measurements from Beromünster will provide invaluable input for ongoing modeling studies aiming to quantify regional greenhouse gas fluxes by inverse modelling.

### Acknowledgements

This project is supported by the Swiss National Science Foundation through the Sinergia program, CarboCount CH project (CRSII 2-136273). We thank Rüdiger Schanda for looking after the technical installations at the Beromünster site and Hanspeter Moret, Peter Nyfeler, and the mechanical workshop team for manufacturing the inlet- and valve switching units.



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



## List of Figures

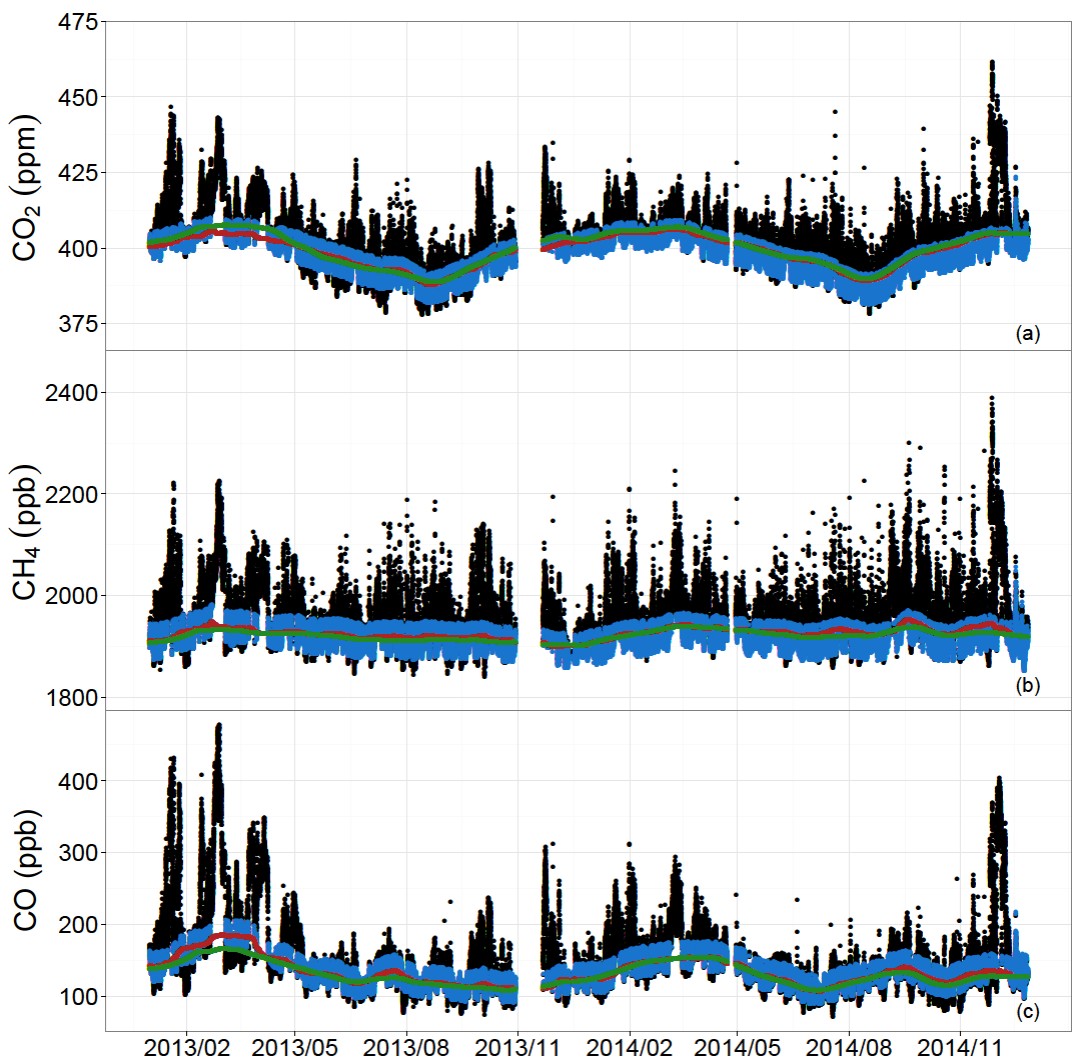

**Figure 1** Time series (December 2012-December 2014) of $CO_2$ (a), $CH_4$ (b), and CO (c) at Beromünster for air sampled at 212.5 m. Color codes indicate: all measurements (black), estimated background values using 2-$\sigma$ filter (blue), moving average of 30 days based on the background estimates (red), and seasonal variations estimated by REBS (green).



**Figure 2** Slope and $r^2$ for $\Delta CO/\Delta CO_2$ (a, b) and $\Delta CO/\Delta CH_4$ (c, d) for the sampling levels 12.5 m (black), 44.6 m (blue), 71.5 m (green), 131.6 m (red) and 212.5 m (yellow). Horizontal dashed lines represent Swiss inventory estimates for 2013.





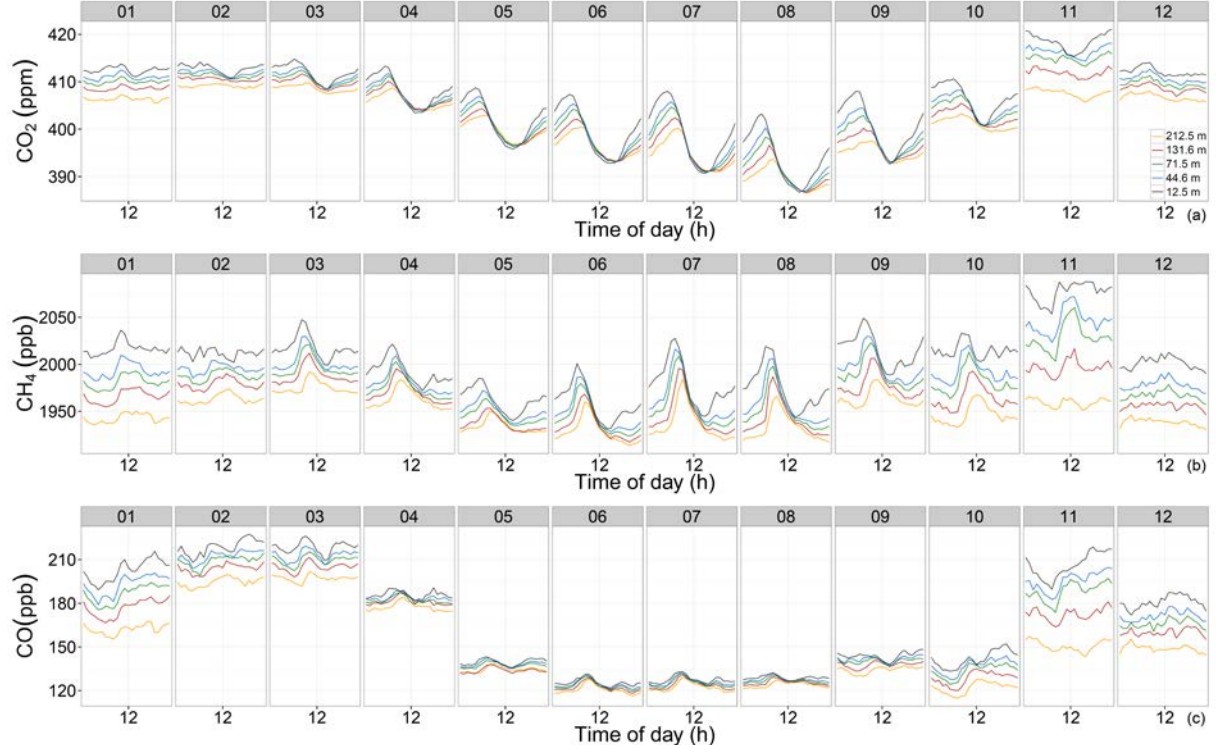

**Figure 3** Monthly mean diurnal cycles of $CO_2$ (a), $CH_4$ (b) and CO (c) for the sampling levels 12.5 m (black), 44.6 m (blue), 71.5 m (green) 131.6 m (red), and 212.5 m (yellow). The x axis on each subplot is centered at noon (UTC 12:00).





**Figure 4** Mean diurnal cycles of $CO_2$ (a), $CH_4$ (b), CO (c) in June for the sampling levels 12.5 m (black), 44.6 m (blue), 71.5 m (green) 131.6 m (red), and 212.5 m (yellow) together with the cosine of the solar zenith angle ($\cos\theta_s$) calculated for the geographical coordinates of site Beromünster.





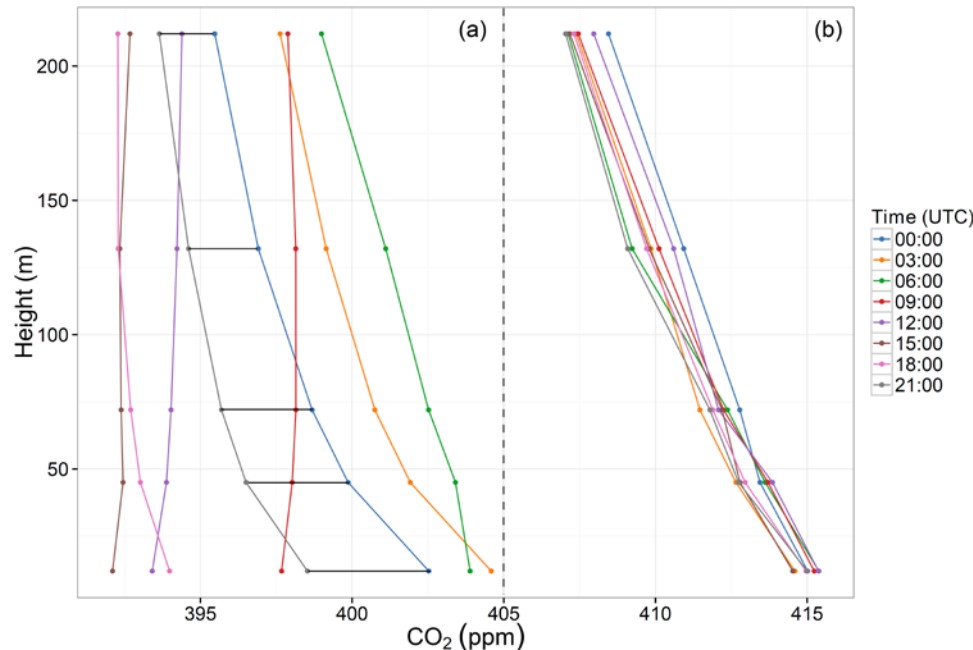

**Figure 5** Development of monthly mean $CO_2$ profile by time of day along Beromünster tall tower (between 12.5 and 212.5 m) during June 2013 (a) and January 2013 (b).



**Figure 6** Diurnal cycle of the storage flux estimates for $CO_2$ (a), $CH_4$ (b) and CO (c) for the years 2013 (red) and 2014 (blue).





**List of Tables**

**Table 1** Calculated growth rates from 2013-2014, seasonal amplitudes, and storage fluxes of species measured at Beromünster. The numbers in brackets are the estimates using the REBS technique.

| Species | Growth rate ($yr^{-1}$) | Amplitude | Storage flux (g C/$m^2$) | |
|---|---|---|---|---|
| | | | 2013 | 2014 |
| $CO_2$ (ppm) | $1.78 \pm 0.05$ ($1.80 \pm 0.01$) | 13.1 | -29 | -35 |
| $CH_4$ (ppb) | $9.66 \pm 0.37$ ($9.69 \pm 0.07$) | - | | |
| CO (ppb) | $-1.27 \pm 0.26$ ($-0.20 \pm 0.09$) | 47.7 | - | - |

