# Peer review of "Continuous CO2/CH4/CO measurements (2012-2014) at Beromünster tall tower station in Switzerland"

_Biogeosciences, 2015_

## Referee Comment (RC1) · Anonymous Referee #2 · 26 Jan 2016

The authors instrumented a radio tower in Beromünster (Switzerland) for measurement of atmospheric CO2, CH4, and CO mixing ratios at 4 different height levels up to 212.5m above ground. They analyze the first 2 years of high-frequency measurements with respect to growth rate, seasonality, mean diurnal cycles, and tracer-tracer correlations. From the vertical profiles, they calculate storage fluxes as a proxy of local surface sources/sinks. All the results are compared to other tall tower measurements and to information about local/regional tracer fluxes, and discussed in terms of their implications for the processes causing the observed trace gas variations.

I find this an important and interesting work, adding information to better understand the highly complex cycles of greenhouse gases in Europe. In addition to the information on the local/regional trace gas processes, the setting up and operation of this tall tower site is a very valuable contribution to the continental and word-wide observation network.
[Figure]

Its value will even increase further with continuation of the measurements.

The paper is very well and clearly written. I like the concise language and the way the authors put their work into context with other observations and the European greenhouse gas cycles. I clearly recommend this work for publication in Biogeosciences. Except for very few very minor suggestions below, I have no comments to add.

Minor comments:

p2 line 20: remove spurious "were"

p3 line 14: clarify the measurement schedule by adding ". . .was conducted **successively** . . ." (if that is what you did).

p4 line 21: It seems that "For the estimation. . ." starts a new topic, which could be clarified by a new paragraph.

p5 lines 6-15: I feel it would be good to make the meaning of the storage flux clearer. If I understood the meaning correctly, it might suffice to add in lines 14-15 "**surface** source" and "**surface** sink".

p8 line 6: you probably mean singular "maximum"

p12 lines 16-19: Mention where the Winderlich et al measurements were done.

---

## Referee Comment (RC2) · Anonymous Referee #1 · 14 Feb 2016

The paper is describing the first two years of measurements of atmospheric mole fractions of CO2, CH4 and CO at Beromünster, Switzerland. The time series are analyzed to characterize seasonal, diurnal and correlation between species. Since this is a new monitoring site, I do recommend the paper for publication if the following points are considered: - the accuracy and repeatability of the measurements must be developed, even if this is more developed in another submitted manuscript. It is important to know the precision of the measurement, and how it has been assessed; - similarly the storage flux estimates are lacking an evaluation of the uncertainties, making very difficult to know what are the significant results. Only very vague comments are done about the uncertainties. - the authors must add information about the data availability, and where the time series can be downloaded.

Measurement system: I understand that the measurement protocol is fully describe in

another manuscript currently under discussion, however I would request the authors to add information about 1/ the calibration of the instrument, and 2/ the assessment of repeatability and accuracy, and how it is controlled.

Seasonal variations: Background vs local/regional signals: I found confusing the presentation of the pollution events due to local and regional sources/sinks. You have to make it clear what do you consider as a 'regional' event. High values observed in winter during hours to days most probably correspond to frontal system passing over Western Europe. I am pretty sure that those events could be observed in other background stations in Netherlands, Germany, France or Italy. To me such events correspond to the transport of pollution events at the European scale by synoptic processes. Then in the discussion I noted some inconsistencies, like for example on Page 6 where you explained that after eliminating the so-called local/regional events you attribute the spring maximum observed for CH4 to the agriculture source in Switzerland. To my understanding, Switzerland influence is part of the regional contribution, and so should not influence the background signal. Could you please have a much more clear definition and interpretation of local, regional, background scales?

I am surprised that you do not use any information about wind speed/direction to interpret and possibly classify the dataset. Don't' you have such meteorological observations ?

Page7 (Line12/14): The explanation for having lower minimum at the topmost level of the tower compared to the surface does not convince me. The last sentence: "During summer and spring months when photosynthesis is active and vertical mixing is strong, atmospheric CO2 accumulates near the surface", definitively needs more explanation, especially by making a clear distinction between daytime and nighttime signals.

Correlations between species Page 9 (line 4): Do you have confirmation of the wintertime CO maximum emission by the inventories ?

An analysis of the species correlations with back-trajectories would be useful, for ex-

ample to check if you measure different CO/CO2 ratios for air masses originating from Germany or France.

I agree with your analysis of the dependence of correlations slopes and r2 to the sampling level. I would suggest adding a comment about the CO/CO2 correlation in summertime which is more dependent to the height, which you can probably explain by the decoupling of CO/CO2 sources due to the biosphere activity at this season.

Diurnal variations Could you please remind in this section which dataset you are using for the analysis of diurnal variabilities? I suppose you are using the full dataset without local/regional events filtering.

Page 10 (Line 11): Your interpretation of the CH4 vertical gradients (higher mixing ratios near the surface => strong local sources) seems quite logical. However, looking at figure 4 we see that CO has exactly the same gradient, shifted in time by a couple of hours. Would you explain similarly this gradient by local CO emissions ? Could you please clarify?

Page 10 (Line 18): You mention a pronounced diurnal cycle of CO during summer months, which appears to be in contradiction with the first comment ('diurnal variations not visible') of this paragraph and figure 3c. I guess the vertical scale of the figure doesn't allow seeing the diurnal variations.

I would expect a specific comment for November which seems quite different from other months, with stronger vertical gradients for all species.

Flux estimation

Page 11 (Line18): large uncertainties: can you please provide an estimation of those uncertainties? The figure 6 is difficult to interpret not knowing which signal is significant or not. If none conclusion can be drawn for winter months, it is probably not useful to show those values on figure 6.

Page 11 (Line 23): 1.57 +/- 0.11 : Could you precise what means the +/-0.1. Is it a standard deviation of daily estimates ? Why those values are much lower than Winderlich 2014 results (page 12) ?

Page 12 (Line 7): Would the CO2/CH4 analogy could be used to estimate uncertainties on CO2 fluxes ?

Conclusions Could you please give indications on the data availability, where can they be uploaded, and the status of the tall tower regarding the continuation of the measurements for long term monitoring ?

―――――――――――――――――――

---

## Author Response (AR1)

**Reply to the review of Anonymous Referee #1**

The authors would like to thank anonymous referee #1 for the valuable comments. In the following, referee's comments are given in bold and author's responses in plain text. Suggested new text is quoted in italics together with page and line numbers.

**The paper is describing the first two years of measurements of atmospheric mole fractions of CO2, CH4 and CO at Beromünster, Switzerland. The time series are analyzed to characterize seasonal, diurnal and correlation between species. Since this is a new monitoring site, I do recommend the paper for publication if the following points are considered: - the accuracy and repeatability of the measurements must be developed, even if this is more developed in another submitted manuscript. It is important to know the precision of the measurement, and how it has been assessed; - similarly the storage flux estimates are lacking an evaluation of the uncertainties, making very difficult to know what are the significant results. Only very vague comments are done about the uncertainties. - the authors must add information about the data availability, and where the time series can be downloaded.**

Our replies to individual points raised by the reviewer will be addressed below.

**Measurement system: I understand that the measurement protocol is fully describe in another manuscript currently under discussion, however I would request the authors to add information about 1/ the calibration of the instrument, and 2/ the assessment of repeatability and accuracy, and how it is controlled.**

We do understand the reviewers concerns that more information on calibration procedures and data reliability should be given.

Section 2.1 will be extended as follows (page 3 line 17):

> *"In order to calibrate the ambient air measurements, standard gases bracketing the expected ambient mixing ratios were measured once a week. In addition, a working standard was measured every 6 hours to monitor the measurement drift and a target gas once a day to check the accuracy and long term stability of the system. From the calibrated target gas measurements a long-term reproducibility of 2.79 ppb, 0.05 ppm and 0.29 ppb for CO, $CO_2$ and $CH_4$ was calculated over the 19 months of the measurements. The overall accuracy has been estimated as 3.48 ppb, 0.07 ppm and 0.30 ppb for CO, $CO_2$ and $CH_4$ measurements respectively (for details see Berhanu et al., 2015)."*

**Seasonal variations: Background vs local/regional signals: I found confusing the presentation of the pollution events due to local and regional sources/sinks. You have to make it clear what do you consider as a 'regional' event. High values observed in winter during hours to days most probably correspond to frontal system passing over Western Europe. I am pretty sure that those events could be observed in other background stations in Netherlands, Germany, France or Italy. To me such events correspond to the transport of pollution events at the European scale by synoptic processes.**

The reviewer is right that the term regional is not well defined. It should be considered to be somewhere in between the local (site) scale and continental scale, but certainly larger than a region of Switzerland. Oney et al. (2015) analyzed the spatial representativeness of the four

CarboCount-CH sites. Based on Lagrangian particle dispersion simulations, the authors estimated that 50% of the total regional surface influence is contained within a distance of about 200 km from the tower. The remaining influence from outside this region is amplified towards the west due to predominantly westerly winds (Oney et al., 2015).

The reviewer is certainly right that a lot of these variations are due to synoptic scale variability, notably the passage of fronts. These events provide important information on regional (sub-continental) fluxes since these changing meteorological conditions lead to variations in the spatial "sampling" of sources and sinks over Europe both due to changes in air mass origin and vertical mixing.

See below for the proposed changes to the manuscript.

**Then in the discussion I noted some inconsistencies, like for example on Page 6 where you explained that after eliminating the so-called local/regional events you attribute the spring maximum observed for CH4 to the agriculture source in Switzerland. To my understanding, Switzerland influence is part of the regional contribution, and so should not influence the background signal. Could you please have a much more clear definition and interpretation of local, regional, background scales?**

We never meant to attribute the springtime maximum to the agricultural source in Switzerland. We will rearrange the sentences to make clear that the two statements are not directly connected.

We agree that the terms "background", "regional" and "local" influence should have been defined more clearly. In the paper, background and baseline are used interchangeably for the filtered time series. "Background" is not a precisely defined term but it is generally considered to represent the spatially smoothly and temporally slowly varying portion of the measured signal characterizing air masses that have not been influenced by recent emissions (or losses). Background concentrations can best be measured at remote sites like Mace Head, Ireland, but are more challenging to determine at more polluted sites like Beromünster.

In order to be clear with the definitions of local, regional and background, we propose to make following changes:

On page 3 (line 23):

> *"For the analysis of seasonality, first we estimated a smoothly varying signal, which we refer hereafter as baseline or background, based on the measurements at the highest elevation using the complete 25-month record. This background is considered to represent the concentrations that would have been observed if the air mass had not been influenced by recent emissions during its transport over the European continent."*

And on page 5 (line 20):

> *"Points outside of the blue band are composed of either pollution or depletion events due to local (<10 km) and regional (some hundreds of km) sources and sinks and are often related to synoptic variability of atmospheric transport and mixing."*

On page 6 (line 20), reformulation:

> *"The rather constant CH₄ values at Beromünster in summer indicate that the separation into background and polluted air is not perfect and that our baseline contains a non-negligible contribution from regional emissions which obscures the expected summertime minimum."*

**I am surprised that you do not use any information about wind speed/direction to interpret and possibly classify the dataset. Don't' you have such meteorological observations?**

We indeed have meteorological observations as stated on page 3 (line 17). They have been analyzed in the publication by Oney et al. (2015) to characterize and compare the local meteorological conditions at all four CarboCount-CH sites. Local wind conditions will likely correlate with the measurements to some extent, but they are not a reliable indicator of air mass origin and surface influence (as could be determined with an atmospheric transport model like FLEXPART) and thus are not very well suited to classify the data. The analysis presented in Oney et. al (2015) shows that winds are strongly channeled between the Alps and the Jura mountains along a south-west to north-east axis (Fig. 4 in their publication) and that wind speeds have a diurnal cycle with a minimum in the morning (around 9-10 local time) and a maximum around mid-night (at the highest elevation). The latter is likely due to the top of the tower being usually located above the nocturnal boundary layer. Therefore, a classification based on the local wind would not necessarily be very useful and diurnal effects would have to be separated from synoptic effects.

**Page7 (Line12/14): The explanation for having lower minimum at the topmost level of the tower compared to the surface does not convince me. The last sentence: "During summer and spring months when photosynthesis is active and vertical mixing is strong, atmospheric CO2 accumulates near the surface", definitively needs more explanation, especially by making a clear distinction between daytime and nighttime signals.**

We agree that the sentence on page 7 line 12-14 was confusing. For a more comprehensive explanation, the following changes will be included:

> *"This could be explained by the atmospheric rectifier effect: Photosynthesis and thermally driven convective mixing are both regulated by the sun, and therefore show the same variability patterns on seasonal and diurnal time scales. During times of strong convective mixing (daytime), photosynthesis is the dominating process, whereas during times of weak mixing (nighttime), respiration dominates (Denning et al., 1999). Therefore, the time-mean vertical profile of $CO_2$ mixing ratios over vegetation shows higher values near the surface than aloft."*

To further illustrate this, Fig. R1 shows the seasonal variations of $CO_2$ at all heights for the background time series. Here, it is seen that the lowest level (black line) experiences the largest peak-to-peak amplitude and the highest mixing ratios throughout the year. When compared to the other levels, the larger amplitude of the lowest level is related to higher winter maximum rather than lower summer minimum. The lowest summer minimum is observed at the highest level (yellow line).

[Figure]

**Figure R1** Seasonal variations of the time series (December 2012 – December 2014) of $CO_2$ at Beromünster for air sampled at 12.5 m (black), 44.6 m (blue), 71.5 (green), 131.6 m (red) and 212.5 m (yellow)

**Correlations between species Page 9 (line 4): Do you have confirmation of the wintertime CO maximum emission by the inventories?**

Unfortunately there are no reliable time functions for seasonal or diurnal variability for CO, and the inventories only provide data for the annual means. According to the Swiss Federal Office of Environment (2015), the main sources of CO in Switzerland are the transport sector and residential heating (page 6 line 26). Considering that CO is a product of incomplete combustion and large emissions have been attributed to residential wood burning, we expect higher emissions in winter.

**An analysis of the species correlations with back-trajectories would be useful, for example to check if you measure different CO/CO2 ratios for air masses originating from Germany or France.**

The reviewer is right that a back-trajectory analysis would be useful; however within the scope of this publication, we decided to limit ourselves to a purely measurement-based analysis without using models. Further studies, which are currently in preparation, will investigate the $CO/CO_2$ ratios in more detail by including radiocarbon data as well as Lagrangian particle dispersion model simulations.

**I agree with your analysis of the dependence of correlations slopes and r2 to the sampling level. I would suggest adding a comment about the CO/CO2 correlation in summertime which is more dependent to the height, which you can probably explain by the decoupling of CO/CO2 sources due to the biosphere activity at this season.**

This is a good point. However, one should be careful with such interpretation since the coefficients of determination ($r^2$) during the summer months are close to zero and the slopes are therefore not well defined.

**Diurnal variations Could you please remind in this section which dataset you are using for the analysis of diurnal variabilities? I suppose you are using the full dataset without local/regional events filtering.**

Yes, we used the full dataset though the highest and lowest 5% of measurements per month and hour were discarded as stated on page 4 (line 18).

The following sentence will be added on page 9 (line 17):

> "For the calculation of monthly mean diurnal cycles, trimmed datasets were used, in which the highest and lowest 5% of the measurements per month and hour were excluded (see Sect. 2.2)."

**Page 10 (Line 11): Your interpretation of the CH4 vertical gradients (higher mixing ratios near the surface => strong local sources) seems quite logical. However, looking at figure 4 we see that CO has exactly the same gradient, shifted in time by a couple of hours. Would you explain similarly this gradient by local CO emissions? Could you please clarify?**

The reviewer is right about the comment on CO. Indeed, the later peak of the CO emissions might be attributed to the traffic, which shows a distinct diurnal cycle with small night-time emissions and peaks in the morning due to rush hours. Additionally, the later peak might as well be related to transport of emissions from the surrounding valleys.

We modified page 10 (line 26) as:

> "For CO, rather constant nighttime values were observed compared to $CH_4$. This indicates that the source of CO is not from the direct vicinity of the tower ($< 2km$), but transported from the surrounding valleys. The increase in the mixing ratios during the morning coincides with the rush hour, suggesting emissions from the traffic. Therefore a later peak is observed for the CO emissions."

**Page 10 (Line 18): You mention a pronounced diurnal cycle of CO during summer months, which appears to be in contradiction with the first comment ('diurnal variations not visible') of this paragraph and figure 3c. I guess the vertical scale of the figure doesn't allow seeing the diurnal variations.**

The reviewer is certainly right that the vertical scale of Fig. 3c does not allow seeing diurnal variations. CO experiences weaker diurnal cycles compared to its annual cycle. Therefore, Fig. 4c is provided as a zoom in for a summer month (June) to highlight the diurnal cycle.

For clarity the sentence will be rephrased on page 10 (line 14):

> "All species show clear diurnal variations though the diurnal variation of CO (Fig. 3c) is hardly visible since it is much smaller than its annual cycle."

**I would expect a specific comment for November which seems quite different from other months, with stronger vertical gradients for all species.**

Indeed November looks quite different when compared to the months before and after. This could be explained by data availability. In November 2013 (1-21 November), the measurement instrument Picarro CRDS (G-2401) had a failure and was replaced by a Picarro CRDS (G-2311-f) which does not measure CO mixing ratios (Berhanu et al., 2015). For the consistency of the dataset between the measured species, we have not included the data from the second analyzer in our analysis. In Fig. R2, diurnal variations of $CO_2$ and $CH_4$ are shown for 1-21 November 2013 (left panel) and all November data of the years 2013 and 2014 (right panel). Compared to Fig. 3 in the manuscript, the consideration of the missing data in November shifts the mean diurnal variation by a couple of ppm down for $CO_2$ and a few tens of ppb for $CH_4$.

[Figure]

**Figure R2** Diurnal variations of $CO_2$ and $CH_4$, data points from Picarro CRDS G2311-f (left), all data available in November (right) for the sampling levels 12.5 m (black), 44.6 m (blue), 71.5 m (green), 131.6 m(red) and 212.5 m (yellow). The x axis on each subplot is centered at noon (UTC 12.00).

The following sentences will be added on page 3 (after line 18):

> *"Due to a malfunction during November 2013 (1-21 November), the measurement instrument (Picarro CRDS G-2401) was replaced by another instrument (Picarro CRDS G-2311-f) which does not measure CO mixing ratios. For the consistency of the dataset between the measured species we only show measurements from Picarro CRDS G-2401."*

The following sentences will be added on page 10 (after line 21):

> *"In November the diurnal variations of all species show marked differences compared to the months before and after (Fig. 3). This can mostly be explained by data availability. As mentioned in Sect. 2.1, three weeks of measurements are missing in November and the monthly mean diurnal cycles were calculated using a relatively small amount of data."*

**Flux estimation**
**Page 11 (Line18): large uncertainties: can you please provide an estimation of those uncertainties? The figure 6 is difficult to interpret not knowing which signal is significant or not. If none conclusion can be drawn for winter months, it is probably not useful to show those values on figure 6.**

We do agree that Fig. 6 is somewhat difficult to interpret; however, even if wintertime fluxes are not significant it is still useful to show them in order to highlight the differences between the seasons. Therefore, we would like to keep Fig. 6 as it is in the manuscript.
For better visibility, we provide a rearrangement of Fig. 6 as shown in Fig. R3, centering the April- September period where flux estimates are most robust and most interpretations should be done.
As explained in Sect. 2.3 (page 5 line 16/17) in the manuscript calculations are done on a daily basis and then averaged over a month, resulting in monthly averaged daily fluxes. The

uncertainties of the monthly mean hourly fluxes were calculated as the standard error of the mean, considering the number of days as the sample size. For winter months, as stated in page 11 (line 4/5) and shown in Fig. 5b, the concentration increments between the consecutive heights were very low resulting in near zero flux estimates (see Fig. R3). The calculated uncertainties were as large as the flux estimates.

The sentence on page 11 (line 17/18) will be complemented as follows:

> *"Most calculated monthly mean hourly flux estimates for the winter months were insignificantly different from zero for all species and included uncertainties as large as the signals."*

**Page 11 (Line 23): 1.57 +/- 0.11 : Could you precise what means the +/-0.1. Is it a standard deviation of daily estimates? Why those values are much lower than Winderlich 2014 results (page 12)?**

As stated above and on page 5 (line 16/17) in the manuscript, the uncertainties were calculated using the standard error of the mean (i.e., the standard deviation divided by the square root of the number of samples contributing to the mean). The mean of the daily flux estimates during the specified time interval (month and hours) is calculated and its standard error is reported. In Fig. 7 and Fig. 9 Winderlich et al (2014) show the error of the means as well, however, it is not reported how the uncertainties on the mean values of the total nighttime fluxes, as given on page 12 were obtained.

The sentence on page 12 (line 19) will be complemented as follows:

> *"In Beromünster taking the summer months (June – September) yielded a mean storage flux of 1.8 ± 0.2 (standard error) $\mu mol\ m^{-2}s^{-1}$ and 3.6 ± 1.0 (standard error) $nmol\ m^{-2}s^{-1}$ for $CO_2$ and $CH_4$ respectively. In addition to standard error of the means, we report the average of the monthly standard errors as 0.87 $\mu mol\ m^{-2}s^{-1}$ and 6.0 $nmol\ m^{-2}s^{-1}$ for $CO_2$ and $CH_4$ respectively."*

**Page 12 (Line 7): Would the CO2/CH4 analogy could be used to estimate uncertainties on CO2 fluxes?**

Indeed it might be possible to estimate uncertainties on $CO_2$ fluxes using the negative $CH_4$ values in the afternoon. This would correspond to a distribution of the estimated $CO_2$ flux during the afternoon among plant uptake and vertical mixing. However, we would still not be able to overcome the underestimation of the calculated fluxes during the afternoon hours where turbulent mixing is dominant (page 11 line 10/14).

[Figure]

**Figure R3** Diurnal cycle of the storage flux estimates for $CO_2$ (a) $CH_4$ (b) and CO (c) for the years 2013 (red) and 2014 (blue)

**Conclusions Could you please give indications on the data availability, where can they be uploaded, and the status of the tall tower regarding the continuation of the measurements for long term monitoring?**

At the moment, data can be obtained upon request. As agreed among the project participants, they will be made available to the public a couple of months after the official completion of the CarboCount-CH project (end of December 2015). Details will be decided at an upcoming meeting. The measurements are presently continued by the University of Bern, however without having access to long-term funding yet.

**Reply to the review of Anonymous Referee #2**

The authors would like to thank anonymous referee #2 for the positive comments. In the following, referee's comments are given in bold, author's responses in plain text. Suggested new text is quoted in italics together with page and line numbers.

**The authors instrumented a radio tower in Beromünster (Switzerland) for measurement of atmospheric CO2, CH4, and CO mixing ratios at 4 different height levels up to 212.5m above ground. They analyze the first 2 years of high-frequency measurements with respect to growth rate, seasonality, mean diurnal cycles, and tracer-tracer correlations. From the vertical profiles, they calculate storage fluxes as a proxy of local surface sources/sinks. All the results are compared to other tall tower measurements and to information about local/regional tracer fluxes, and discussed in terms of their implications for the processes causing the observed trace gas variations.**

**I find this an important and interesting work, adding information to better understand the highly complex cycles of greenhouse gases in Europe. In addition to the information on the local/regional trace gas processes, the setting up and operation of this tall tower site is a very valuable contribution to the continental and word-wide observation network.**

**Its value will even increase further with continuation of the measurements. The paper is very well and clearly written. I like the concise language and the way the authors put their work into context with other observations and the European greenhouse gas cycles. I clearly recommend this work for publication in Biogeosciences.**

**Except for very few very minor suggestions below, I have no comments to add.**

**Minor comments:**
**p2 line 20: remove spurious "were"**
**p3 line 14: clarify the measurement schedule by adding "...was conducted \*\*successively\*\*..." (if that is what you did).**
**p4 line 21: It seems that "For the estimation ..." starts a new topic, which could be clarified by a new paragraph.**
**p5 lines 6-15: I feel it would be good to make the meaning of the storage flux clearer. If I understood the meaning correctly, it might suffice to add in lines 14-15 "\*\*surface\*\* source" and "\*\*surface\*\* sink".**
**p8 line 6: you probably mean singular "maximum"**

We do agree with the suggestions above, the changes will be made in the manuscript accordingly.

**p12 lines 16-19: Mention where the Winderlich et al measurements were done**

The sentence will be rearranged accordingly, page 12 (line 16):

[revised manuscript text omitted]

---

## Author Response (AR2)

Reply to the Review of Anonymous Referee #1

The authors would like to thank anonymous referee #1 for the valuable comments and the improvements on the earlier version of the manuscript. In the following, referee's comments are given in bold and author's responses in plain text. Suggested new text is quoted in italics together with page and line numbers.

**I agree with all the modifications proposed by the authors. I think the revised version gives a clearer picture of the dataset and the interpretation of the observed signals. I only have three minor points:**

**Measurement system: I think the few lines describing the regular measurement of calibration and target gases is useful. From this short paragraph I understand that the 'overall accuracy' is estimated from the target gas. If this is the case I would not use the term 'overall accuracy' since the target gas measurement will not allow you to take into account the possible bias due to the water vapor correction, or any contamination in the inlet line. Could you please make this clarification in the manuscript.**

The reviewer is right; we do estimate the accuracy using the target gas measurements only.

Therefore the following changes will be made on page 3 (line 21):

> "*The calculated accuracy using the target gas measurements corresponds to 3.48 ppb, 0.07 ppm and 0.30 ppb for CO, $CO_2$ and $CH_4$ measurements respectively (for details see Berhanu et al., 2015). Note that the overall accuracy might be slightly larger due to uncertainties in water vapor correction and potential contamination in the sample inlet lines.*"

**Wind speed/direction: The explanation you gave in your answer for not using the meteorological information to classify the dataset is satisfactory and I think you could insert this argument in the manuscript.**

The following will be added on page 4 (line 21):

> "*Although meteorological observations are done at the tower, they were not used in the baseline selection. Oney et al. (2015) has analyzed the meteorological data in order to characterize and compare the local meteorological conditions at all four CarboCount-CH sites. Local wind conditions will likely correlate with the measurements to some extent, but they are not a reliable indicator of air mass origin and surface influence and thus are not very well suited to classify the data. According to Oney et al. (2015), winds are strongly channeled between the Alps and the Jura mountains along a south-west to north-east axis and wind speeds have a diurnal cycle with a minimum in the morning and a maximum around mid-night (at the highest elevation). The latter is likely due to the top of the tower being usually located above the nocturnal boundary layer. Therefore, a classification based on the local wind would not necessarily be very useful and diurnal effects would have to be separated from synoptic effects.*"

**Conclusions: same point regarding the data availability. It should be included in the manuscript.**

Data availability will be added to the manuscript accordingly (page 14, line 10):

*"Currently, data can be obtained upon request."*

References:

[revised manuscript text omitted]